# Investigating transmission of SARS-CoV-2 using novel face mask sampling: a protocol for an observational prospective study of index cases and their contacts in a congregate setting

Thomas Jaenisch [1,2] Molly M Lamb,[1] Emily N Gallichotte,[3] Brian Adams,[4] Charles Henry,[5] Jeannine Riess,[6] Joni Triantis van Sickle,[6] Kellie L Hawkins,[7] Brian T Montague,[8] Cody Coburn,[8] Leisha Conners Bauer,[9] Jennifer Kovarik,[9] Mark T Hernandez,[10] Amy Bronson,[11] Lucy Graham,[12] Stephanie James,[13] Stephanie Hanenberg,[14] James Kovacs,[15] John S Spencer,[3] Mark Zabel,[3] Philip D Fox,[3] Olivia Pluss,[4] William Windsor,[4] Geoffrey Winstanley,[4] Daniel Olson,[4] Michael Barer,[16] Stephen Berman,[4] Gregory Ebel,[3] May Chu[4]

For numbered affiliations see end of article.

**Correspondence to**
Dr Thomas Jaenisch;
thomas.jaenisch@cuanschutz.edu

## ABSTRACT

**Introduction** This study aims to measure how transmission of SARS-CoV-2 occurs in communities and to identify conditions that lend to increased transmission focusing on congregate situations. We will measure SARS-CoV-2 in exhaled breath of asymptomatic and symptomatic persons using face mask sampling—a non-invasive method for SARS-CoV-2 detection in exhaled air. We aim to detect transmission clusters and identify risk factors for SARS-CoV-2 transmission in presymptomatic, asymptomatic and symptomatic individuals.

**Methods and analysis** In this observational prospective study with daily follow-up, index cases and their respective contacts are identified at each participating institution. Contact definitions are based on Centers for Disease Control and Prevention and local health department guidelines. Participants will wear masks with polyvinyl alcohol test strips adhered to the inside for 2 hours daily. The strips are applied to all masks used over at least 7 days. In addition, self-administered nasal swabs and (optional) finger prick blood samples are performed by participants. Samples are tested by standard PCR protocols and by novel antigen tests.

**Ethics and dissemination** This study was approved by the Colorado Multiple Institutional Review Board and the WHO Ethics Review Committee. From the data generated, we will analyse transmission clusters and risk factors for transmission of SARS-CoV-2 in congregate settings. The kinetics of asymptomatic transmission and the evaluation of non-invasive tools for detection of transmissibility are of crucial importance for the development of more targeted control interventions—and ultimately to assist with keeping congregate settings open that are essential for our social fabric.

**Trial registration number** ClinicalTrials.gov (#NCT05145803).

## STRENGTHS AND LIMITATIONS OF THIS STUDY

⇒ The study employs a novel and non-invasive sampling device.
⇒ Following index cases as well as their contacts prospectively enables us to study transmissibility.
⇒ Results from the novel sampling device in the face masks are compared with standard nose swab sampling.
⇒ The manuscript serves as a template for similar studies to be replicated.

## INTRODUCTION

There is an urgent need to better describe and analyse the transmission of SARS-CoV-2 in asymptomatic and presymptomatic (infected) individuals, especially regarding the heterogeneity of transmissibility ('super-spreader events'), and within the context of congregate situations. An innovative tool with high sensitivity and minimal invasiveness is needed to obtain the necessary information to fill the current knowledge gaps.

A sizeable proportion of individuals infected with SARS-CoV-2 are asymptomatic. These individuals contribute to transmission and are also less likely to get tested. Asymptomatic SARS-CoV-2-infected individuals carry viral loads in conventional samples from the nasopharynx that are in the same range as those of symptomatic individuals, and a significant proportion of SARS-CoV-2 spread is due to asymptomatic individuals.[1–6] An analytical model published by the Centers for Disease Control and Prevention (CDC)

found that up to 59% of all transmissions likely resulted from asymptomatic transmission. The model further delineated that 35% of these 59% of infections resulted from those that eventually became symptomatic, while 24% remained asymptomatic.[7] This study also found that the asymptomatic group positive for SARS-CoV-2 was only 25% less infectious compared with those that were symptomatic. This is in contrast to findings from systematic review studies focused on secondary attack rates where the authors concluded that secondar attack rates caused by asymptomatic individuals were only around 1% (and 7% in presymptomatic)[8] or within household transmission around 3% (asymptomatic) and 20% (presymptomatic).[9] Overall, few studies were designed specifically to study transmission in congregate situations, including index cases and their close contacts.

With the emergence of virus variants with the potential for greater infectivity, there is also a need to monitor viral variants in congregate situations among asymptomatic individuals, and to rapidly provide new information to public health authorities.[10] This is especially true for breakthrough infections that require special monitoring regarding the details of the vaccines (type, intervals, etc) as well as the virus variants that caused the breakthrough infection.[11]

The reference method for detection of SARS-CoV-2 is the reverse transcriptase PCR, a technology that detects viral RNA from respiratory specimens (sputum or nasal swab) or from saliva. For diagnostic purposes, these samples are usually collected at a single point in time.

To study the dynamics of SARS-CoV-2 transmission over time, we investigate a non-invasive face mask sampling (FMS) method, where polyvinyl alcohol (PVA) strips are embedded in masks for the detection of SARS-CoV-2 in exhaled breath (figure 1). This enables us to extend the sampling interval over a time window (hours) and include different activities or risk factors such as singing, shouting, sneezing or coughing as part of the data collection process.

The overall aim of this study is to analyse transmission clusters of SARS-CoV-2 in communities and identify factors

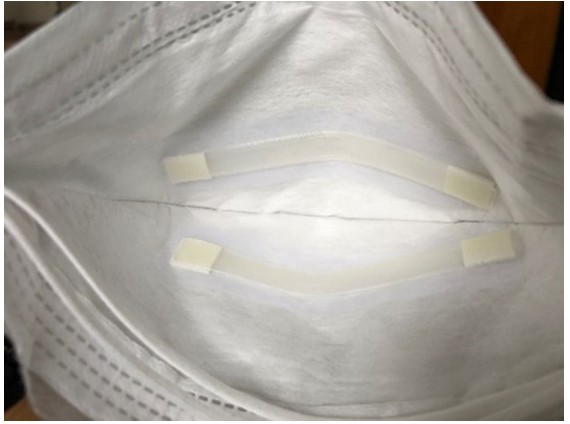

**Figure 1** Two polyvinyl alcohol test strips adhered to the inside of a face mask. Photo credit: Michael Barer.

**Box 1    Objectives**

Study Identifier: clinicaltrial.gov identifier NCT05145803

Objective 1: Identification of risk factors for, and rate of, infection in index cases and for transmission from index case to contacts.

Objective 1.1: Describe demographics and evaluate risk factors for infection in index cases.

Objective 1.2: Evaluate risk factors for transmission from index case to contacts (and their contacts), including viral load, clinical symptoms, vaccination status, activity and environmental metrics and immune response profiles (eg, neutralising antibodies, cytokine profiles—using a blood sample taken after recovery).

Objective 1.3: Analyse the difference in the clinical status (symptomatic vs asymptomatic) for index cases as well as secondary cases by humoral and T-cell immune response profile (using the blood samples collected).

Objective 1.4: Determine rate of and time to transmission, stratified by a variety of risk factors (including presence of clinical symptoms, activity and environmental metrics both in index cases and secondary contacts), analysing presymptomatic and asymptomatic exhalation of SARS-CoV-2 in contacts of index cases.

Objective 2: In confirmed index cases and their contacts, quantify and determine the efficacy of the polyvinyl alcohol (PVA) test strip method for SARS-CoV-2 detection, comparing with the respective 'reference standard' (eg, nasal swab) collection method and correlating with the presence of antibodies.

Objective 2.1: Assess the association between molecular detection by PVA test strip and molecular or antigen detection/quantification by initial reference diagnostic collection method.

Objective 2.2: Assess the association between molecular detection by PVA test strip and molecular or antigen detection method (self-sampling) over the sampling/mask-wearing period.

Objective 3: Due to the concern around emerging variants with higher transmissibility, carry out deep sequencing of PCR-positive samples (swabs as well as strips from masks) to derive the genotype of infecting virus.

Objective 3.1: Share in timely fashion the results of the sequencing with the national/regional surveillance system.

Objective 3.2: Assess the genetic similarities and differences of the SARS-CoV-2 variants by sequence comparison and by virological culture to assess cell-based infectivity of the strains.

Objective 3.3: Determine the chain of transmission and relationship to clinical phenotype in the index case and contacts where viral RNA or culture permit.

Objective 3.4: Analyse the difference in rate of transmission (see also objective 1) comparing different SARS-CoV-2 variants.

that may increase transmissibility, with a focus on congregate situations. We will compare the results from the FMS with the standard nasal swab. Within substudies, we are interested in better understanding antibody profiles and their correlation with symptom status and transmissibility, as well as the effect of different activities (such as singing or exercising) on test positivity in the masks. We also plan follow-up studies (which will need additional ethical approval) investigating long-term outcomes in the college student-age population regarding the incidence and the risk factors associated with long COVID-19. Details of the study objectives can be found in box 1.

| Box 2 Key study design information |
| --- |
| Inclusion/exclusion<br>Inclusion criteria<br>1. Index patients: Individuals aged≥18 years of any gender tested positive for SARS-CoV-2 by a molecular reference test (PCR test or antigen test).<br>2. Contacts: Individuals aged≥18 years and known to be contacts of the index patients.<br>Exclusion criteria: Age<18 years old, unable to wear a face mask due to underlying condition, unable to consent, pregnancy. In addition, signs of severe disease present at time of enrolment (eg, difficulty breathing, pain when breathing, tightness of chest or developing irregular heartbeat).<br>Endpoints<br>A. Outcome measure(s)<br>Objective 1 outcomes: polyvinyl alcohol (PVA) strip positivity; nose swab test positivity.<br>Objective 2 outcomes: PVA strip positivity and virus quantification (by PCR); nose swab test positivity; antigen test positivity; presence of antibodies in the blood.<br>Objective 3 outcome: Deep sequencing of PCR-positive samples to derive the genotype of the infecting virus. |

## METHODS AND ANALYSIS

### Study design

This is a prospective study with daily follow-up to capture early transmission in the presymptomatic phase. Index cases are identified by the routine testing system implemented at each participating institution and contacts of index cases are identified by the site-specific contact tracing system. The contact definition is based on CDC and the local health department guidelines (ie, individuals who are cohabitating or who were in close (<6 feet), extended (>15 min) contact with the index case and within 48 hours of diagnosis) and will be updated to reflect the most recent definition available. Study sites have an ongoing active case detection and contact tracing programme. Inclusion criteria and endpoints are described in box 2.

### Study population

This study started in the first quarter of 2021 and is currently under way in college students in Colorado, USA, including University of Colorado (Denver, Boulder, Colorado Springs, Aurora), Colorado State University (Fort Collins), Regis University (Denver), as well as Colorado Mesa University (Grand Junction). We envisage enrolment to be completed by April 2022 but would potentially continue beyond this if high SARS-CoV-2 transmission is documented. Students as well as faculty (and their family) are eligible to be enrolled.

### The FMS method

The FMS method has been successfully used to detect *Mycobacterium* in breath samples and recently adapted for COVID-19 testing.[12] The sensitivity compared with regular nasal swab sampling has not been conclusively determined (which is part of this research project). The

(preliminary) analytical sensitivity of this method was estimated around ≥$10^4$ virus particles, which corresponds to a cycle threshold (Ct) value of 18–20 in the PCR and is in agreement with recent publications and sample testing by other colleagues at the University of Alberta, Canada.[13]

### Face mask selection

Commercial three-layer spunbond polypropylene masks will be provided to the participants, tested for breathability (95% filtration;KN95). Masks were approved by the occupational safety and health administration (OSHA) / food and drug administration (FDA). Each mask will be uniquely coded for tracking.

### Participant and public involvement

Participants were involved in the selection of best fitting masks in a pilot phase. Feedback was elicited regarding the acceptability and fit of different mask designs.

### PVA test strips

PVA test strips are provided by University of Leicester under a material transfer agreement and strips will be adhered to predetermined positions inside the face masks. PVA has a very good safety profile, being already used in cosmetic products (face masque) and is approved as hypoallergenic and non-toxic for humans.[14] PVA has been used in diagnostic tests[15–17] as well as medical device implant for orthopaedic surgery[18] and for theranostic nanoplatform therapeutic imagery.[19]

### Sample size

This is an exploratory study. We aim for an up to 3300 individuals, including participants with confirmed SARS-CoV2 infection (index cases) and their contacts. These estimates are based on the conservative estimate of 5% transmission from index cases to a close contact. With a maximum of 300 index cases and ~10 contacts per index case, we estimate that we would detect ~150 secondary cases in contacts.

### Schedule of wearing masks

Participants will be asked to wear a mask with two PVA test strips adhered to the inside surface for 2 hours per day. Index cases will complete 7 days of mask wearing and if they test positive by the end of the 7-day period, they will be asked to continue for another 7 days up to day 14. Contacts will complete 7 days of mask wearing as long as they test negative. If testing positive in the first 7 days, they will complete an additional 7 days of mask wearing starting with the day when they first test positive. If by the end of this period they still test positive, the duration will be extended by 2-day intervals with their permission until two consecutive masks wear dates test negative (figure 2). On completion of 2 hours of wear by the participant, the face masks will be placed in a sealed plastic bag, collected daily and brought into the laboratory where the PVA strips will be put into tubes with viral transport medium (VTM). The VTM tubes with the PVA strips will be eluted and tested by PCR.

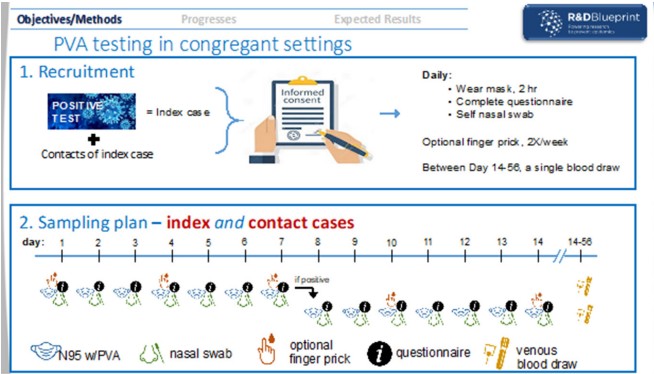

**Figure 2** Polyvinyl alcohol (PVA) mask strip study, sampling plan for index case and contact.

### Self-administered nasal swabs

We ask the participants to collect a self-administered nasal swab daily by placing a nasal swab approximately 0.5 inches into one nostril and swabbing the inside of the nose. The swab is then immediately placed into a 15 mL conical tube containing 2 mL of VTM. Samples are put into a separate sealed bag and stored at 4°C until testing.

### Questionnaires

An electronic questionnaire is administered at the beginning of the mask-wearing period to collect basic demographics (age, gender) and to assess COVID-19 symptoms and activities (including the day when symptoms started). During the follow-up period, daily questionnaires are administered, and we have built in warning signs for potential severe COVID-19 that trigger an auto-alert, in which case we will proactively contact the participant and (a) ask the participant to stop wearing the masks and (b) advise the participant to seek medical care immediately. We also included an optional assessment of activities such as singing or exercising for index cases only (confirmed SARS-CoV-2 positive individuals) on 1 day using a separate second mask. In the future, the assessment of vital signs or specific symptoms could be added via wearable technologies or app-based recording (see, eg, https://www.hyfeapp.com/).

### Blood sampling

We will ask the participants to conduct self-sampling of blood using dried blood spots from finger pricks[20] every third day (optional—filter papers and self-sealing bags provided along with the daily masks). Antibodies will be eluted from the filter paper to be used for serological investigations. These samples are going to be stored in separate sealed bags and picked up together with the other materials. After recovery, for example, between day 14 and 56 after quarantine/self-isolation, the participants are invited to return to the clinic for a venous blood sample (EDTA [Ethylenediaminetetraacetic acid] 20–30 mL). The blood is collected to investigate humoral, and T-cell based immune responses including cytokine profiles and other T-cell phenotypic surface markers

that may be associated with transmissibility and clinical phenotype (eg, protection vs long COVID-19).

### Laboratory methods

In a biosafety cabinet, PVA strips are removed from inside of the face masks, by cutting them at the ends where they are adhered using double-sided tape. Each strip is placed into a 15 mL conical tube containing 2 mL of viral transport medium (VTM) and incubated at room temperature for at least 30 min to allow PVA to dissolve. Samples are stored at 4°C until testing. After collection and elution of the strip, testing will be carried out with the following tests in order to validate the results from the FMS against the reference test:

1. First line: PCR from the eluate of the strip.
2. Second line: antigen test, which gives a positive/negative/indeterminate response.
3. Third line: virus culture to determine if virus is infectious.

In brief, for PCR testing, two approaches are used. In the first approach, samples (PVA strips and nasal swabs) are briefly vortexed to thoroughly mix. Viral RNA is extracted from 200 µL of sample using the Mag-Bind Viral DNA/RNA 96 Kit following the manufacturer's protocol. Quantitative real-time PCR is performed using CDC 2019-nCoV primer probes and EXPRESS qPCR Supermix and SuperScript. Samples are considered positive if their N1 Ct is less than 38. In the second approach, samples (PVA strips and nasal swabs) are briefly vortexed to thoroughly mix. Viral RNA is extracted from 45 µL of sample by digesting with 5 µL of Proteinase K (PK, NEB) for 15 min at 60°C. PK is inactivated for 5 min at 95°C. Reverse transcription droplet digital PCR (RT-ddPCR) was performed on PK digested sample using CDC 2019-nCoV primer probes and One-Step RT-ddPCR Advanced Kit for Probes (Bio Rad). Samples are considered positive if their N1 concentration is >0.25 copies/µL. The RNA extracted from the FMS and potentially recovered live viruses will be used as basis for deep sequencing.[3]

One of our aims is to validate a novel antigen assay for testing PVA strips from the masks. For this assay, electrodes were prepared according to Samper *et al*.[21] For sample analysis, 20 µL of PVA mask sample or virus standard was added to the functionalised electrode and incubated for 40 min, and then rinsed with PBST (phosphate buffer saline with tween) and PBS (phosphate buffer saline) using solid stream spray bottles. Virus standards were inactivated with 0.1% Igepal and diluted in Hanks' Balanced Buffer Solution with 0.1% Tween80 and 0.1% Igepal. Following sample incubation, 20 µL of a 0.5 µg/mL horseradish peroxidase-antibody solution was added and incubated for 25 min, and then rinsed with PBST and PBS using solid stream spray bottles. For measurement, 50 µL of 3,3′,5,5′-tetramethylbenzidine was added to the electrode and incubated for 2 min. After 2 min, chronoamperometry was run at 0.0 V for 2 min.

## Contactless delivery and pick-up of supplies

All masks and other samples are delivered/picked up at the residence of the participants and placed in their own sealed bag with a unique identifier. All exposed masks and collected sample kits are individually packaged compliant with UN3314 packaging safety requirements. For the decontamination of surfaces and personal protective equipment, we will use 70% alcohol for immediate decontamination of exposed surfaces, especially those that the participant may have touched. For longer-term decontamination, we will use 10 µM methylene blue spray that allows for up to 10 hours of decontamination in light.[22] Hand hygiene solutions will be used to clean hands at the end of each drop-off/pick up activity.

## Data management

All study data will be collected using Research Electronic Data Capture (REDCap). Data cleaning and quality control assessments will be conducted on the REDCap survey data, and a weekly data dashboard will be produced to support the study team in detecting issues with data collection. Indicators of severe disease will be monitored daily. Test results from the reference test, the additional molecular tests, antibody tests and the PVA strip test will be linked to survey data using a unique anonymized study identification number, and data will be stored on a secure internal server with restricted access at the University of Colorado with no personal identifiers. The study analyst will hold the key to the personal identifiers and will unblind results as necessary to inform the institution of any previously unknown positive cases, so that they can follow the institution's protocols regarding informing the case, contact tracing and quarantine. All data will be cleaned, merged and analysed using SAS V.9.4 (Cary, North Carolina, USA), to ensure replicability of the process and analysis.

## Data analysis

For objective 1, we will first conduct a descriptive analysis of the index cases. Then we will conduct a risk factor analysis using multivariable multilevel regression models to determine symptoms and behaviours that are associated with greater exhalation of SARS-CoV-2 virus, as detected by PVA test strips. We will also analyse presence and timing of symptoms in contacts to determine if PVA strips can detect SARS-CoV-2 in exhaled breath of presymptomatic and asymptomatic individuals. We will map the transmission dynamics visually, determine the length of exhalation of SARS-CoV-2 in cases and calculate the difference in rate of transmission by vaccination status. Using the convalescent blood sample, we will analyse different immune response profiles (neutralising antibodies, T-cell-based assays) stratified by clinical phenotype (symptomatic/asymptomatic, potentially using a symptom score for increased granularity).

To address objective 2, we will calculate the sensitivity and specificity of the FMS method, using the initial reference diagnostic method of the site in index cases. We will then calculate the sensitivity and specificity of the PVA test strips using the concurrently collected nasal swab as the reference standard for both index cases and their contacts. We will also correlate strip test positivity with Ct value of the PCR test (or viral load in a subset) over time and with presence of antibodies (IgA and IgG) over time.

To address the concern about emergence of new virus variants, in objective 3, the FMS eluted materials, original specimen and/or recovered live virus isolates will be subjected to deep sequencing to monitor genetic variants in the study population. We will use these data, when available, to calculate the infectiousness of each viral variant and to map the transmission chains and association with disease in contacts. We will provide the sequence information of identified SARS-CoV-2 variants (and aliquot of material) to national and global data warehouses for public health surveillance of virus variation that may impact disease transmission and resistance to vaccines. The remaining materials will be stored for longer term so that they can be accessed for public health surveillance purposes.

In addition, the samples and data generated in this study will provide ample opportunity for laboratory analysis such as immune profiling or viral genome copy quantification and sequencing. Specific analyses will depend on the experiments and number of confirmed cases, however, will likely include ANOVA (analysis of variance), student t-test and other analyses. Laboratory data will be examined for normality of distribution as necessary, and the appropriate parametric or non-parametric descriptive analyses will be conducted to determine significant differences.

## Ethics and dissemination

Ethical approval for a multicentric study in adults (college students and faculty) was obtained by the Colorado Multiple Institutional Review Board (COMIRB) at Anschutz Medical Campus in Aurora (#21-2823) and by the WHO Ethics Review Committee (CERC.0096). In addition, ethical approval for a pilot study in middle schools (not described in detail in this manuscript) was obtained by the COMIRB (#21-4944) and an adapted protocol for elementary schools is prepared. Here, we will concentrate on the study in the adult population.

There is only minimal risk to the participant associated with this study. Discomfort may arise form wearing masks for extended periods of time or from additional reference validation tests (optional, nasal swabs, sputum tests and/or blood samples), though these are routine procedures for SARS-CoV-2 protection, testing and surveillance in the partnering sites/ institutions. For validation purposes, participants will

also carry out self-sampling of the anterior nose, which is a minimal risk procedure, and does not cause a lot of inconvenience (different from the nasopharyngeal sampling). For comparison purposes, participants may also be asked to provide a saliva sample, which is not associated with any risk and minimal inconvenience. One venous blood sample (20–30 mL) will be taken after recovery to investigate humoral, and T-cell-based immune responses, including cytokine profiles and other T-cell phenotypic surface markers that may be associated with transmissibility and clinical phenotype (protection vs long-term COVID-19). This is a minimal risk procedure, carried out by professionals. Inconvenience includes possible bruising and some pain. Finger prick blood samples are optional and are considered minimal risk. They might be inconvenient, and we will let participants decide about participating in this.

We present this study design as a tool to further investigate the transmissibility of SARS-CoV-2. We want to share our scientific approach implemented in institutes of higher education which have routine COVID-19 monitoring in place. We would be happy to help replicating this study in other settings worldwide. This study will help determine transmission chains and risk factors for transmission of SARS-CoV-2 in congregate settings. PVA test strips, a non-invasive method, are investigated as a novel and promising tool to detect exhalation of SARS-CoV-2 from people who are actively infected as well as their exposed contacts.

**Author affiliations**
[1] Center for Global Health and Department of Epidemiology, Colorado School of Public Health, Aurora, Colorado, USA
[2] Heidelberg Institute of Global Health, University Hospital Heidelberg, Heidelberg, Germany
[3] Department of Microbiology, Immunology, and Pathology, Colorado State University, Fort Collins, Colorado, USA
[4] Center for Global Health, Colorado School of Public Health, Aurora, Colorado, USA
[5] Department of Chemical and Biological Engineering, Colorado State University, Fort Collins, Colorado, USA
[6] Office of Environmental Health Services, Colorado State University, Fort Collins, Colorado, USA
[7] Public Health Institute, Denver Health, Denver, Colorado, USA
[8] Occupational Health and Division of Infectious Diseases, University of Colorado School of Medicine, Denver, Colorado, USA
[9] Health Promotion and Collegiate Recovery Center, University of Colorado Boulder, Boulder, Colorado, USA
[10] Civil, Environmental and Architectural Engineering, University of Colorado Boulder, Boulder, Colorado, USA
[11] Office of the Vice President, Colorado Mesa University, Grand Junction, Colorado, USA
[12] Department of Health Sciences, Colorado Mesa University, Grand Junction, Colorado, USA
[13] Department of Pharmaceutical Sciences, Regis University, Denver, Colorado, USA
[14] Wellness Center, University of Colorado Colorado Springs, Colorado Springs, Colorado, USA
[15] Department of Chemistry and Biology, University of Colorado Colorado Springs, Colorado Springs, Colorado, USA
[16] Department of Infectious Diseases, University of Leicester, Leicester, UK

**Contributors** TJ, MML and MC conceptualised the study. MC acquired funding. MB developed the face mask sampling method. MML and WW developed the data base, curate the data and developed the data analysis approach. MC, GE, EG, CH, MH, JSS, MZ, JKovacs and PF carry out lab investigations and developed the laboratory testing methodology. SB, TJ, GW, BA, KLH, JTvS, JR, BM, CC, LCB, JKovarik, AB, LG, SJ, SH, OP, WW and DO developed the participant recruitment strategy and conduct the recruitment. TJ and MC wrote the first draft. All authors contributed to the writing of the manuscript and approved the final version.

**Funding** The authors acknowledge that funds for the project were provided by WHO based on a grant from the German Federal Ministry of Health (BMG). In addition, the study was supported in part by Colorado State University's Office of the Vice President for Research's Accelerating Innovations in Pandemic Disease initiative through support from The Anschutz Foundation (to JSS).

**Competing interests** TJ, MML, WW, BA, GE and MC report grant support from WHO for this study for salaries or equipment/reagents. SB, OP, AB, CH, JR, JKovacs and MZ report grant/salary support for unrelated research without conflict of interest. EG, MH, CC, DO, SH, JKovarik, LCB, JTvS, SJ, JSS and LG report no conflict of interest. MZ reports speaker honoraria at academic institutions. BM reports industry support from Regeneron and Eli Lilly Foundation (as investigator, to the institution), MB reports support from the UK National Core Study on Transmission, and PF reports stock in Darwin Biosciences and support (equipment and reagents) from Ceres Nanosciences.

**Patient and public involvement** Patients and/or the public were involved in the design, or conduct, or reporting, or dissemination plans of this research. Refer to the Methods and analysis section for further details.

**Patient consent for publication** Not applicable.

**Provenance and peer review** Not commissioned; externally peer reviewed.

**ORCID iD**
Thomas Jaenisch http://orcid.org/0000-0002-9464-571X

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
