## [Reviewer comments · BMJ Open]

ARTICLE DETAILS

TITLE (PROVISIONAL)	INVESTIGATING TRANSMISSION OF SARS-COV-2 USING NOVEL FACE MASK SAMPLING: A PROTOCOL FOR AN OBSERVATIONAL PROSPECTIVE STUDY OF INDEX CASES AND THEIR CONTACTS IN A CONGREGATE SETTING
AUTHORS	Jaenisch, Thomas; Lamb, Molly; Gallichotte, Emily; Adams, Brian; Henry, Charles; Riess, Jeannine; van Sickle, Joni Triantis; Hawkins, Kellie; Montague, Brian; Coburn, Cody; Connors Bauer, Leisha; Kovarik, Jennifer; Hernandez, Mark; Bronson, Amy; Graham, Lucy; James, Stephanie; Hanenberg, Stephanie; Kovacs, James; Spencer, John; Zabel, Mark; Fox, Philip; Pluss, Olivia; Windsor, William; Winstanley, Geoffrey; Olson, Daniel; Barer, Michael; Berman, Stephen; Ebel, Gregory; Chu, May

VERSION 1 – REVIEW

REVIEWER	Ortiz-Prado, Esteban Universidad de Las Américas
REVIEW RETURNED	24-Mar-2022

GENERAL COMMENTS	The protocol presented by the authors is interesting. Definitely the fact of knowing the transmission in crowded places and what is the potential role of respirator wearers in 95 and whether the symptomatological status is of significant importance.
--

REVIEWER	Kindrachuk, Jason University of Manitoba
REVIEW RETURNED	20-Apr-2022

GENERAL COMMENTS	Jaenisch and colleagues provide a study protocol for a prospective investigation of SARS-CoV-2 transmission using face mask sampling within congregate settings. It is noted that this protocol was reviewed by the Colorado multiple institutions review board and the WHO ethics review committee. There were no major concerns noted with study design. However, some minor concerns were identified: -it was noted that the sensitivity of the authors' testing method was equal or greater than 10^4 viral particles. The authors should qualify how this matches with recent estimates of viral loads from breakthrough infections for Delta and Omicron as reported by Puhach et al: https://www.nature.com/articles/s41591-022-01816-0 -The authors did not note any statistical analysis for laboratory analyses (eg. immune profiling, viral genome copy quantification, etc). -on page 8, lines 55-57, has the time component defining extended contact been adapted with the increased transmissibilities associated with VOCs?
---

	-on page 5, lines 11-16, there is some discussion on the role of asymptomatic infections in SARS-CoV-2 transmission. These modelling values should be contrasted with those from systematic reviews reported by Madewell et al (https://jamanetwork.com/journals/jamanetworkopen/fullarticle/2783544) and Qiu et al (https://pubmed.ncbi.nlm.nih.gov/33484843/). -the authors have chosen to use congregant to describe the settings in their study. A minor point but is this more frequently reported as congregate in the literature?
--	---

REVIEWER	Korzen, Marcin West Pomeranian University of Technology in Szczecin, Department of Artificial Intelligence and Applied Mathematics
REVIEW RETURNED	04-Jul-2022

GENERAL COMMENTS	The paper describes the proposition of future experiments or an initial research stage. I have no objections to the proposed procedures. Research seems to be planned properly and may be very interesting. However, these are only plans without any results and firm conclusions. I do not know how it is seen in this journal, but I think that most scientists expect that one paper presents the closed experiment as a whole in a compact form. In my opinion, here may be the case of a redundant publication, and also the STROBE Statement is incomplete.
---

VERSION 1 – AUTHOR RESPONSE

Reviewer: 1

Dr. Esteban Ortiz-Prado, Universidad de Las Américas, Universitat de Barcelona, Comments to the Author:

The protocol presented by the authors is interesting.

Definitely the fact of knowing the transmission in crowded places and what is the potential role of respirator wearers in 95 and whether the symptomatological status is of significant importance.

RESPONSE: We thank the reviewer for this encouraging evaluation.

Reviewer: 2

Prof. Jason Kindrachuk, University of Manitoba, Comments to the Author:

Jaenisch and colleagues provide a study protocol for a prospective investigation of SARS-CoV-2 transmission using face mask sampling within congregate settings. It is noted that this protocol was reviewed by the Colorado multiple institutions review board and the WHO ethics review committee. There were no major concerns noted with study design. However, some minor concerns were identified:

-it was noted that the sensitivity of the authors' testing method was equal or greater than 10^4 viral particles. The authors should qualify how this matches with recent estimates of viral loads from breakthrough infections for Delta and Omicron as reported by Puhach et al:

<https://www.nature.com/articles/s41591-022-01816-0>

RESPONSE: The recovery of virus was related to the sensitivity of the tissue culture cells and the virus strain. Our Inoculating virus was from Feb 2020, and it did not replicate in high titer as later virus

variants, therefore the detection limit was equal or greater than 10^4 viral particles as reported. (see also the companion manuscript in AJIC reporting first results: <https://doi.org/10.1016/j.ajic.2022.01.010>)

-The authors did not note any statistical analysis for laboratory analyses (eg. immune profiling, viral genome copy quantification, etc).

RESPONSE: The main objectives of the study are centered around the epidemiology transmission. We have included more information on the planned analytical approach for laboratory analyses:

“In addition, the samples and data generated in this study will provide ample opportunity for laboratory analysis such as immune profiling or viral genome copy quantification and sequencing. Specific analyses will depend on the experiments and number of confirmed cases - however, will likely include ANOVA, student t-test, and other analyses. Laboratory data will be examined for normality of distribution as necessary, and the appropriate parametric or non-parametric descriptive analyses will be conducted to determine significant differences.”

-on page 8, lines 55-57, has the time component defining extended contact been adapted with the increased transmissibilities associated with VOCs?

RESPONSE: Thanks for this interesting question. We had initially defined this according to the CDC definition – and have now checked the latest guidelines by CDC, which still states that the definition of ‘close contact’ is determined by two factors: a) being less than 6 feet (1.5m) apart, b) for a cumulative total of 15 minutes over 24 hours. If necessary, we would adapt the time component according to new CDC guidelines and have included a sentence accordingly:

“... and will be updated to reflect the most recent definition available.”

-on page 5, lines 11-16, there is some discussion on the role of asymptomatic infections in SARS-CoV-2 transmission. These modelling values should be contrasted with those from systematic reviews reported by Madewell et al (<https://jamanetwork.com/journals/jamanetworkopen/fullarticle/2783544>) and Qiu et al (<https://pubmed.ncbi.nlm.nih.gov/33484843/>).

RESPONSE: Thanks, we have included and discussed these new references. The relevant paragraph now reads:

“This is in contrast to findings from systematic review studies focused on secondary attack rates where the authors concluded that secondary attack rates caused by asymptomatic individuals were only around 1% (and 7% in pre-symptomatic) {Liu et al., 2021} or within household transmission around 3% (asymptomatic) and 20% (pre-symptomatic) {Madewell et al., 2021}. Overall, few studies were designed specifically to study transmission in congregant situations, including index cases and their close contacts.”

-the authors have chosen to use congregant to describe the settings in their study. A minor point but is this more frequently reported as congregant in the literature?

RESPONSE: Thanks for this point. We have replaced ‘congregant’ with ‘congregate’.

Reviewer: 3

Dr. Marcin Korzen, West Pomeranian University of Technology in Szczecin Comments to the Author:

The paper describes the proposition of future experiments or an initial research stage. I have no objections to the proposed procedures. Research seems to be planned properly and may be very interesting. However, these are only plans without any results and firm conclusions. I do not know how it is seen in this journal, but I think that most scientists expect that one paper presents the closed experiment as a whole in a compact form. In my opinion, here may be the case of a redundant publication, and also the STROBE Statement is incomplete.

RESPONSE: We have written this manuscript as a study protocol manuscript, which per definition does not include any results. We checked and understood that BMJ ONE does publish study design/protocol manuscripts. This also explains why the STROBE statement is incomplete. We also refer to the document "Note from the Editors: Instructions for reviewers of study protocols", where it is specified that "While some baseline data can be presented, there should be no results or conclusions present in the study protocol."

Reviewer: 1

Competing interests of Reviewer: None to declare

Reviewer: 2

Competing interests of Reviewer: None

Reviewer: 3

Competing interests of Reviewer: None.

VERSION 2 – REVIEW

REVIEWER	Kindrachuk, Jason University of Manitoba
REVIEW RETURNED	31-Aug-2022
GENERAL COMMENTS	There are no additional concerns that need to be addressed by the authors.